# Demonstration of an always-on exchange-only spin qubit

Joseph D. Broz [1] ✉, Jesse C. Hoke [1], Edwin Acuna [1] & Jason R. Petta [1,2,3] ✉

In conventional exchange-only (EO) spin qubit demonstrations, quantum gates have been implemented using sequences of individually pulsed pairwise exchange interactions with only one exchange coupling active at a time. Alternatively, multiple non-commuting exchange interactions can be pulsed simultaneously, reducing circuit depths and providing protection against leakage. We demonstrate high-fidelity quantum control of an always-on exchange-only (AEON) qubit, operated using simultaneous exchange pulses in a triangular quantum dot (QD) array. We use blind randomized benchmarking to characterize the performance of the full AEON single-qubit Clifford gate set, achieving an average Clifford gate fidelity $F_{C1} = 99.86\%$. Extensions of this work may enable more efficient EO two-qubit entangling gates as well as the implementation of native $i$-Toffoli gates in Loss-DiVincenzo single-spin qubits.

Gate-defined semiconductor QDs are an attractive platform for scalable quantum computing owing to their small size, compatibility with existing semiconductor fabrication techniques, and potential for operation at temperatures above 1 K[1–7]. In QDs, quantum information is typically stored in the spins of confined electrons or holes, which benefit from long coherence times[8,9] and the availability of a fast, electrically controlled exchange interaction for coupling nearest neighbor spins[10,11]. However, single-spin qubit control is difficult, requiring either precise engineering of local magnetic fields or the use of materials with strong spin-orbit coupling, both of which add to design complexity[12–14].

The EO qubit was developed to circumvent the need for single-spin control by encoding a single qubit in a subspace of three physical spins, which support universal quantum control through the exchange interaction alone[15]. The encoded gates are typically implemented using sequences of pulsed pairwise exchange interactions, where no more than two spins interact at any given time[16–19]. This serial mode of operation is chosen for its practical simplicity, as qubit control is characterized by a single parameter—a time-integrated exchange energy $J_{i,j}(t)$ coupling spins $i$ and $j$—which is easy to tune and calibrate[18]. However, this simplicity comes at the cost of large gate depths, with recently demonstrated single-qubit gates requiring up to four exchange pulses and two-qubit gates as many as twenty-eight[18,19]. EO gates can be designed to further mitigate leakage error sources, but at the cost of even longer sequence lengths, such as the 16-pulse single-

qubit gates designed to dynamically decouple magnetic and charge noise[20], or the 44-pulse leakage-controlled two-qubit gates which suppress the spreading of leakage errors[19].

The AEON qubit is a variant of the EO qubit that uses simultaneously pulsed pairs of exchange couplings to construct quantum gates[21]. A key advantage of the AEON qubit is significantly shorter gate depths, with single-qubit gates requiring no more than two simultaneous exchange pulses[22] and two-qubit gates as little as one[23,24]. Moreover, these gates are naturally protected from leakage due to an induced energy gap between the qubit and leakage subspaces[23–25].

Here we report full control of an AEON qubit in a Si/SiGe triangular QD array[26]. Sensitivity to charge noise is mitigated by operating exchange at a double sweet spot (DSS), which is first-order insensitive to fluctuations of the chemical potentials of all three confined electrons[21,27]. We implement a calibration procedure for AEON qubit gates that simultaneously calibrates both the axis and angle of rotation. With this procedure, we realize a complete set of gates and extract a single-qubit Clifford gate fidelity $F_{C1} = 99.86\%$ via blind randomized benchmarking (BRB). Our results show that AEON qubit performance is on par with state-of-the-art EO qubits operated using sequential exchange pulses[18].

## Results

Measurements are performed on a triangular triple QD device consisting of a single layer of etch-defined gate electrodes fabricated on

[1]HRL Laboratories, LLC, Malibu, CA, USA. [2]Department of Physics and Astronomy, University of California, Los Angeles, CA, USA. [3]Center for Quantum Science and Engineering, University of California, Los Angeles, CA, USA. ✉e-mail: jdbroz@hrl.com; jpetta1@hrl.com

top of an isotopically enriched $^{28}$Si/SiGe heterostructure (800 ppm $^{29}$Si)[3,26]. A scanning electron microscope image of a nominally identical device is shown in Fig. 1a. Voltages applied to plunger gates ($P_i$) form QDs in the underlying $^{28}$Si quantum well, each tuned to confine a single electron. Voltages applied to exchange gates ($X_{i,j}$) control the exchange coupling between neighboring QDs. In practice, devices are tuned using virtual gates, linear combinations of physical gate voltages that selectively control key dot parameters while compensating for device cross-capacitances[28–30]. Specifically, the virtual plunger gate voltage $\widetilde{V}_{P_i}$ controls the chemical potential $\epsilon_i$ of dot $i$ and the virtual exchange gate voltage $\widetilde{V}_{X_{i,j}}$ controls the tunnel coupling $t_{i,j}$ between dots $i$ and $j$ (see Methods for more details). The remaining gate electrodes are used to load electrons into the dots from charge reservoirs (B, $T_i$) and to form a dot charge sensor (M, $Z_i$)[31].

The Hamiltonian for the three singly-occupied exchange-coupled QDs is given by:

$$\hat{H}(t) = J_{1,2}(t)\hat{S}_2 \cdot \hat{S}_2 + J_{2,3}(t)\hat{S}_2 \cdot \hat{S}_3 + J_{1,3}(t)\hat{S}_1 \cdot \hat{S}_3, \quad (1)$$

where $\hat{S}_i$ are the dot spin operators and we set $\hbar = 1$ such that $J_{i,j}$ can be interpreted as either an energy or angular frequency. We introduce the notation $k$-$J$ to describe the number $k$ of nonzero $J_{i,j}$ contained in $\hat{H}$. EO or AEON qubit control is then distinguished by the restriction of $\hat{H}$ to 1-$J$ or 2-$J$ exchange, respectively. That is, an AEON qubit is distinguished from an EO qubit by its mode of operation. EO qubit control consists of serial pulsing of exchange where only a single exchange interaction is active at a time, while AEON control consists of simultaneous exchange where at least two exchange interactions are active at a time (2-$J$ exchange).

Both the EO and AEON qubit are encoded identically in the collective three-electron spin state of the array, which occupies an eight-dimensional Hilbert space[15]. In terms of the total (three-electron) spin $S$ and its projection $m_S$, this Hilbert space decomposes into a $S = 3/2$ quadruplet ($m_S = \pm 1/2, \pm 3/2$) and two $S = 1/2$ doublets ($m_S = \pm 1/2$). The exchange interaction, which conserves both $S$ and $m_S$, only couples states within the two doublets. The qubit is encoded by defining $|0\rangle = |S_{13} = 0, S = 1/2, m_S\rangle$ and $|1\rangle = |S_{13} = 1, S = 1/2, m_S\rangle$, where $S_{13}$ is the combined spin of dots 1 and 3 and $m_S = \pm 1/2$ acts as an extra gauge degree of freedom[1]. The state $|0\rangle$ corresponds to a singlet between dots 1 and 3, with the uncoupled spin of dot 2 determining the qubit's gauge. We use standard Pauli spin blockade techniques[11] to initialize the qubit into the singlet state $|0\rangle$ and to map the occupation of $|0\rangle$ onto the charge configuration, which can then be measured using the dot charge sensor[31]. The specific value of $m_S$ is randomly assigned during initialization and is left unresolved by measurement.

Expressed in the qubit basis, Eq. (1) takes the form:

$$\hat{H}(t) = -\frac{1}{2}\left[\sqrt{3}J_-(t)\hat{\sigma}_x + (J_{1,3}(t) - J_+(t))\hat{\sigma}_z\right], \quad (2)$$

where $\hat{\sigma}_i$ are the standard Pauli operators, and we define $J_+ = (J_{1,2} + J_{2,3})/2$ and $J_- = (J_{1,2} - J_{2,3})/2$. A detailed derivation of Eq. (2) from Eq. (1) is provided in the Supplemental Material of ref. 18. In the Bloch sphere picture, time-evolution under $\hat{H}$ corresponds to a rotation with angle $\theta$ about an axis $\hat{r} = (\cos(\varphi), 0, \sin(\varphi))$ in the $xz$-plane, as described by the unitary operator

$$\hat{R}_\varphi(\theta) = \mathcal{T}\exp\left[-i\int_0^\tau \hat{H}(t)dt\right] \quad (3)$$

$$\approx \cos(\theta/2) - i\sin(\theta/2)[r_x\hat{\sigma}_x + r_z\hat{\sigma}_z], \quad (4)$$

where $\mathcal{T}$ is the time-ordering operator, $\theta = \int_0^\tau \Omega(t)dt$, $\Omega = \sqrt{3J_-^2 + (J_{1,3} - J_+)^2}$, $r_x = \cos(\varphi) = \sqrt{3}J_-/\Omega$, and $r_z = \sin(\varphi) = (J_{1,3} - J_+)/\Omega$. The approximation in Eq. (4) is exact only when $[\hat{H}(t), \hat{H}(t')] = 0$ for all times $t$ and $t'$, which requires that the ratios of nonzero $J_{i,j}$ remain constant over the duration of the exchange pulses. This condition is trivially met for 1-$J$ exchange, which drives rotations about one of three non-orthogonal axes separated by 120° in the $xz$-plane [see Fig. 1b]. We label these as $\hat{z} = (0, 0, 1)$, $\hat{n} = -(\sqrt{3}, 0, 1)/2$, and $\hat{m} = (\sqrt{3}, 0, -1)/2$. By interleaving rotations about any two of these axes, an arbitrary single-qubit gate can be constructed using at most four 1-$J$ exchange pulses[18].

In contrast, 2-$J$ exchange allows for rotations about an arbitrary axis in the $xz$-plane [see Fig. 1c]. Using 2-$J$ exchange, any single-qubit gate can be implemented with no more than two pulses[22]. Because the two constituent interactions are non-commuting, 2-$J$ qubit operation generally requires full temporal control of the voltage waveforms. However, we verify through experiment that even without implementing this level of control Eq. (4) still provides an accurate description of 2-$J$ operation.

Due to its encoding in a Zeeman doublet, the qubit's dynamics are invariant to global magnetic fields[15,32–34]. However, fluctuating magnetic field gradients generated by nearby spinful nuclei can induce decoherence and leakage out of the qubit subspace[35,36]. We mitigate leakage by applying a small global magnetic field $B \approx 3$ mT to suppress transitions that do not conserve $m_S$, leaving a single (gauge-dependent) leakage state, $|1, 3/2, m_S\rangle$, spin-degenerate with the qubit

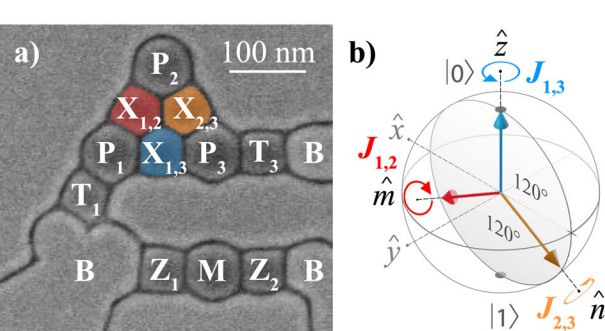
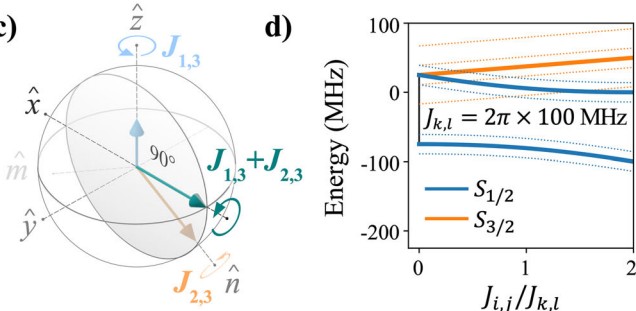

**Fig. 1 | Device operation. a** Scanning electron microscope image of a triangular QD similar to the one measured. **b** During conventional EO qubit operation, a single exchange coupling generates a rotation about one of three non-orthogonal axes ($\hat{z}$, $\hat{n}$, or $\hat{m}$). **c** More efficient control sequences can be implemented if two exchange couplings are simultaneously activated. For example, an $x$-axis rotation can be implemented by applying simultaneous voltage pulses to the $X_{1,3}$ and $X_{2,3}$ gates. **d** Eigenergies of Eq. (1) plotted as a function of $J_{i,j}$ with $J_{k,l} = 2\pi \times 100$ MHz ($i,j \neq k, l$),

$J_{i,l} = 0$, and $B = 0$. Solid blue curves correspond to the $S = 1/2$ doublets and the solid orange curve to the $S = 3/2$ leakage space quadruplets. When both $J_{i,j}$ and $J_{k,l}$ are nonzero, the $S = 1/2$ and $S = 3/2$ subspaces are energetically isolated, suppressing leakage out of the qubit subspace. A nonzero global magnetic field (dotted energy levels), lifts most of the remaining degeneracies. To preserve leakage protection in the presence of a global magnetic field, simultaneous exchange energies should be chosen to avoid level crossings.

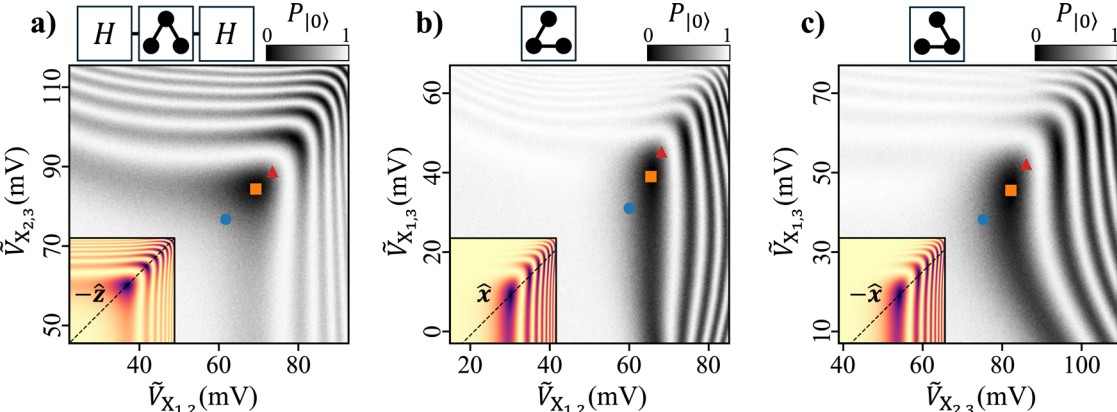

**Fig. 2 | Quantum control with simultaneous exchange couplings. a–c** The probability $P_{|0\rangle}$ of measuring the encoded $|0\rangle$ state following a 10 ns 2-$J$ exchange pulse as a function of pulse amplitude for all pairwise combinations of virtual exchange gates. The colored markers identify the locations of $\pi/2$ (blue circles), $\pi$ (orange squares), and $3\pi/2$ (red triangles) rotations. The target axes about which these rotations are performed are the same as those labeled in the insets. For the data in (**a**), 1-$J$ Hadamard ($H$) rotations are applied before and after the 2-$J$ exchange pulse. Insets: Simulations of the ideal two-level system evolution as given by Eq. (4) and assuming an exponential dependence of the exchange energy on virtual exchange gate voltages.

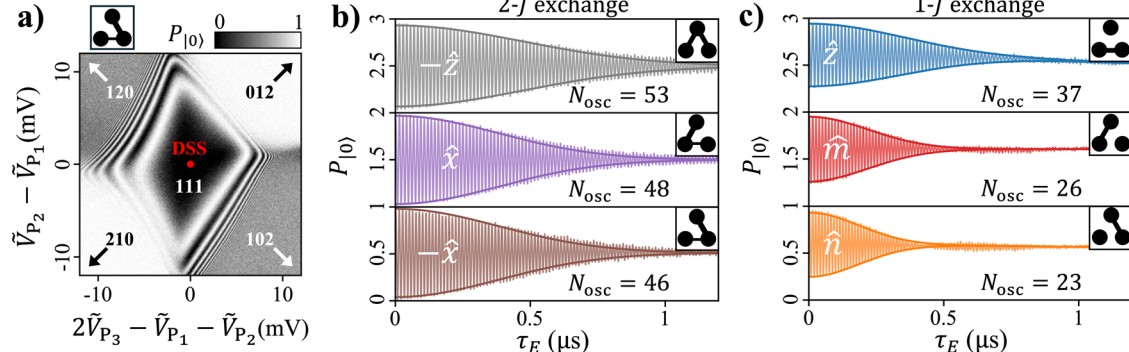

**Fig. 3 | Coherent control of the AEON qubit. a** The 2-$J$ DSS, where the dependence of the exchange energy on dot chemical potentials vanishes to first order. By performing plunger gate sweeps, we identify the DSS as the point in bias space where $P_{|0\rangle}$ exhibits minimal sensitivity to P gate voltages (red circle). The linear combination of plunger gate voltages swept along each axis are chosen to be mutually orthogonal with each other and the common mode voltage $\tilde{V}_{P_1} + \tilde{V}_{P_2} + \tilde{V}_{P_3}$. Tuples denote the locations of (or directions towards) the nearest relevant charge configurations (e.g. 111 denotes the equilibrium state where a single electron is confined in each dot). **b, c** Measured 2- and 1-$J$ exchange oscillations with the oscillation frequency, $\Omega$, tuned to $\sim 2\pi \times 80$ MHz. The white labels denote the targeted axis of rotation and successive curves are offset by 1 on the $y$-axis for clarity. $N_{osc}$ is extracted by fitting each data set to a Gaussian decay envelope (darker lines). We note that pre/post $\pi$-pulses are applied about the $\hat{n}$-axis when performing 1-$J$ $\hat{z}$-rotations (blue curve) and pre/post Hadamard rotations are applied when performing 2-$J$ $\hat{z}$-rotations (gray curve). Insets: Weighted solid lines indicate relative exchange gate pulse amplitudes.

subspace. 1-$J$ exchange partially breaks this remaining degeneracy by introducing an energy gap between states with $S_{13} = 0$ and $S_{13} = 1$. On the other hand, 2-$J$ exchange energetically separates the $S = 1/2$ qubit and the $S = 3/2$ subspaces, highly suppressing leakage [see Fig. 1d].

To characterize the 2-$J$ exchange landscape, we sweep the amplitudes of two virtual exchange gate voltages applied simultaneously during a 10 ns pulse, producing the fingerpinch plots shown in Fig. 2[37]. The data reveal oscillations of the probability $P_{|0\rangle}$ of measuring the encoded $|0\rangle$ state as a function of exchange gate voltages, as expected from Eq. (4). The insets in Fig. 2 show simulations of the corresponding ideal two-level system evolution, assuming an uncoupled exponential dependence of exchange energy $J_{i,j}$ on the virtual exchange gate voltage $\tilde{V}_{X_{i,j}}$. While the measured data qualitatively resembles the simulations, nonlinear and, in some cases, non-monotonic deviations are evident due to a complicated cross-dependence of the 2-$J$ exchange interaction on gate voltages[38].

In principle, a 2-$J$ exchange interaction depends on both the interdot tunnel couplings and the dot chemical potentials. In a triple-dot system, the chemical potentials are typically parameterized using a common mode and two differential modes[39]. Exchange is insensitive to the common mode, defined as $\bar{\epsilon} = \epsilon_1 + \epsilon_2 + \epsilon_3$, but sensitive to the differential modes: the tilt detuning, $\epsilon_t = (\epsilon_2 - \epsilon_1)/2$, and the dimple detuning, $\epsilon_d = \epsilon_3 - (\epsilon_1 + \epsilon_2)/2$. However, it is possible to operate 2-$J$ exchange at a DSS where the sensitivity to the two differential modes vanishes to first order, as demonstrated in Fig. 3a, thereby reducing decoherence due to charge noise[21,27,39–41]. Although exchange energies retain a second-order dependence on chemical potentials at the DSS[21], these contributions are expected to be negligible compared to the dominant first-order dependence on the interdot tunnel couplings, as is the case for 1-$J$ exchange operated at the analogous symmetric operating point[41].

Operating at the DSS, we characterize coherence by measuring the qubit's time-evolution during 2-$J$ exchange pulses, as plotted in Fig. 3b. We quantify coherence by the number of exchange oscillations $N_{osc}$ that occur before the amplitude decays to $1/e$ of its initial value. For comparison, we also plot the measured time-evolution for 1-$J$ exchange operated at symmetric operating points in Fig. 3c (see Supplementary Information for 1-$J$ fingerprint plots and associated

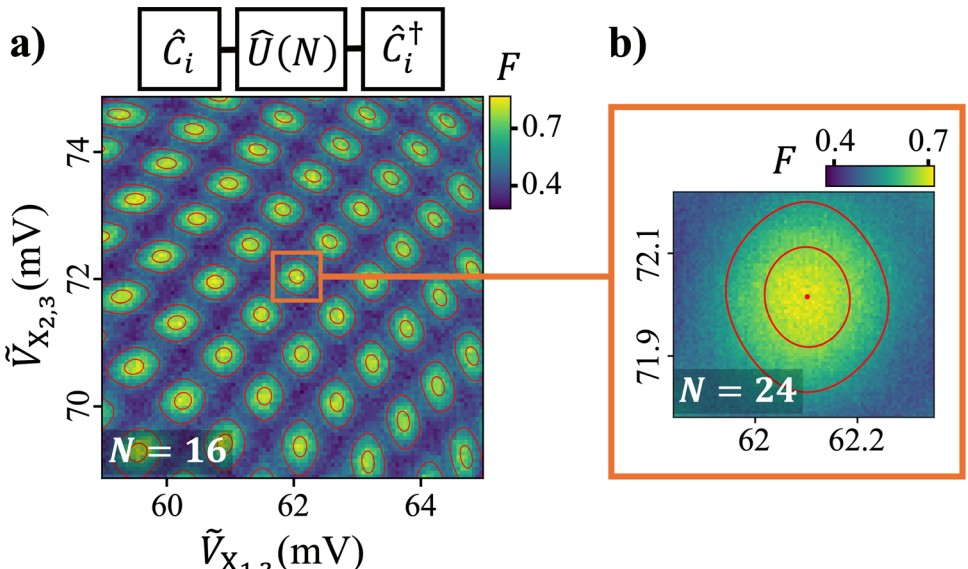

**Fig. 4 | 2-*J* gate calibration. a** *F* obtained by twirling *U* over the 1-*J* single-qubit Cliffords as a function of the amplitudes $\tilde{V}_{X_{1,2}}$ and $\tilde{V}_{X_{2,3}}$ of a 2-*J* exchange pulse. Here, $\hat{U}$ is designed to calibrate a 2-*J* π-rotation about the $-\hat{z}$ axis. The calibration is optimized by choosing the pulse amplitudes that maximize the central peak of *F*

(enclosed by the orange square). **b** Zoom-in of the optimal peak with *N* = 24. The red contour lines are obtained by fitting the data to determine the peak location (see Supplementary Information).

operating points). We find that the $N_{osc}$ for 2-*J* exchange is generally greater than for 1-*J* exchange, despite both regimes operating at sweet spots with vanishing first-order sensitivity to chemical potential fluctuations. More detailed device modeling may be helpful for understanding the coherence enhancement during 2-*J* operation.

To calibrate a 2-*J* exchange rotation $\hat{R}_\varphi(\theta)$, both the rotation axis $\varphi$ and angle $\theta$ must be tuned. We directly optimize the fidelity of a composite pulse sequence designed to be simultaneously sensitive to both 2-*J* rotation parameters. The composite sequence is defined as $\hat{U}(N) = \hat{U}_{ax}^N \hat{U}_{ang}^N$ and reduces to the identity $\hat{\mathbb{1}}$ when the 2-*J* rotation is perfectly calibrated. This occurs when $\hat{R}_\varphi(\theta) = \hat{R}_{\varphi^*}(\theta^*)$ for the target rotation angles $\varphi^*$ and $\theta^*$. The two components of $\hat{U}$ differ in their sensitivity: $\hat{U}_{ax}$ is primarily sensitive to errors in $\varphi$ and $\hat{U}_{ang}$ to errors in $\theta$. Explicitly, we define $\hat{U}_{ang} = \hat{R}_\varphi^{2q}(\theta)$ and $\hat{U}_{ax} = [\hat{R}_\eta(\chi)\hat{R}_\varphi^q(\theta)]^2$, where $q$ is a positive integer chosen such that $q\theta = s\pi$ for some odd integer $s$, and $\hat{R}_\eta(\chi)$ is a pre-calibrated rotation with $\chi = \pi$ and $\eta = \varphi^* \pm \pi/2$. Similar sequences are used in the context of robust phase estimation and gate set tomography to amplify deviations from target gate parameters[42,43]. Finally, to map the fidelity $F(\hat{U}, \hat{\mathbb{1}})$ of $\hat{U}$ relative to the identity onto the measured probability $P_{|0\rangle}$, we twirl $\hat{U}$ over the set of 1-*J* single-qubit Clifford gates[44].

A specific rotation $\hat{R}_\varphi(\theta)$ is calibrated by sweeping the amplitudes of the exchange gate pulses to produce a two-dimensional map of *F*. An example sweep for a 2-*J* π-rotation about the $-\hat{z}$-axis is shown in Fig. 4, where a pre-calibrated 2-*J* π-rotation about the $\hat{x}$-axis is used as $\hat{R}_\eta(\chi)$ in the construction of $\hat{U}_{ax}$. As the exchange voltages are varied, *F* oscillates periodically, producing a series of interference peaks whose frequency—and therefore sensitivity—increases with the sequence length *N*. Only the central peak corresponds to the optimal calibration condition $\varphi = \varphi^*$ and $\theta = \theta^*$, which we track by performing successive sweeps with increasing values of *N* = 1, 2, 4, ..., continuing until reductions in the signal-to-noise ratio prevents scaling beyond *N* = 24. This calibration procedure is effectively a two-dimensional extension of the robust phase estimation protocol of Kimmel et al.[42] and similarly benefits from a Heisenberg scaling with respect to sequence length *N*,

making it well suited for larger multi-qubit systems. In practice, we perform low-resolution sweeps for intermediate values of *N* and use a generic peak-finding algorithm to locate the central maximum, reserving a high-resolution sweep and analytic fit (see supplementary information) for the final *N* = 24 iteration. Using this procedure, a full calibration of a single 2-*J* rotation takes approximately five minutes.

Using this calibration procedure, we tune nine distinct 2-*J* exchange pulses corresponding to rotations of π/2, π, and 3π/2 about each of the axes: $-\hat{z}$, $-\hat{x}$, and $\hat{x}$. The locations of these pulses in bias space are indicated by the colored markers in Fig. 2. To evaluate the performance of each pulse, we perform interleaved BRB, which involves interleaving a calibrated 2-*J* rotation into BRB sequences constructed from 1-*J* exchange pulses[18]. Details on the calibration procedure and interleaved BRB measurements are given in the Supplementary Information.

To quantitatively assess the AEON qubit performance, we execute BRB on 2-*J* single-qubit Clifford gates constructed from the best-performing 2-*J* pulses[18]. In Fig. 5, we compare 2-*J* gate fidelities to the BRB performance of standard 1-*J* EO single-qubit gates in the same device. Our measurements yield a 2-*J* single qubit Clifford gate fidelity $F_{Cl} = 99.86\%$, slightly surpassing the best 1-*J* BRB fidelity of 99.84% obtained from rotations about the $\hat{n}$ and $\hat{z}$ axes. Additionally, the 2-*J* leakage error per Clifford, 0.015%, is approximately half that of the average 1-*J* leakage error per Clifford, 0.029%, with the 1-*J* leakage error being comparable to the value measured in similar devices[3,19,26]. This reduction is consistent with the leakage-protected nature of the 2-*J* gates, as each Clifford gate spends roughly half of its duration at idle, where exchange is negligible.

Although 2-*J* BRB outperforms 1-*J* BRB, this advantage primarily arises from its shorter gate depths: a 2-*J* single-qubit Clifford gate requires an average of 1.9 exchange pulses, compared to 2.7 pulses for a 1-*J* Clifford gate. When this difference is taken into account, the total 2-*J* error per exchange pulse ($\varepsilon_{pp} \approx 0.076\%$) is in fact higher than that of the best-performing 1-*J* gates ($\varepsilon_{pp} \approx 0.060\%$). This outcome is somewhat unexpected given the larger $N_{osc}$ and lower leakage rates observed for 2-*J* exchange, which imply a reduced sensitivity to charge and hyperfine noise—the dominant noise sources in EO qubits[18,19]. This discrepancy suggests an increased contribution from coherent errors

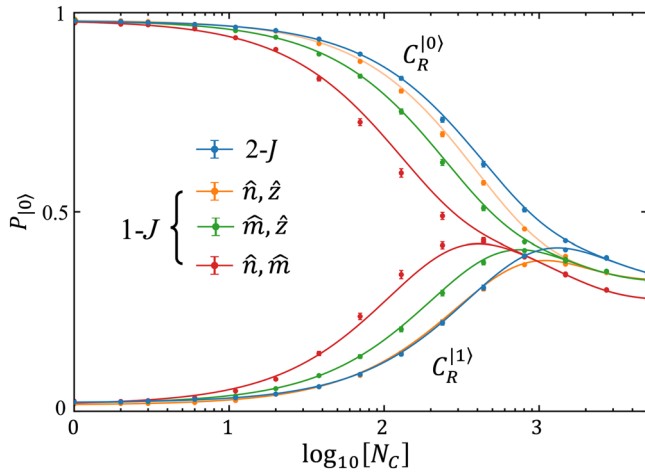

**Fig. 5 | Blind randomized benchmarking.** The blue curve shows the results of BRB performed using a Clifford gate set constructed from 2-$J$ exchange pulses. The average single-qubit Clifford gate fidelity is $F_{C1} = 99.86\%$, and the average leakage rate is 0.015% per Clifford. For comparison, we also perform standard 1-$J$ BRB for all pairwise combinations of 1-$J$ exchange axes (blue, green and red curves). In general, 2-$J$ BRB outperforms 1-$J$ BRB, achieving both a higher average Clifford gate fidelity and a lower leakage rate. Error bars indicate the standard error of the mean probability from 250 sequence repetitions.

during 2-$J$ pulsing, possibly due to imperfect calibration or enhanced sensitivity to pulse overlap[18], but we suspect the dominant contribution stems from the non-commutativity of 2-$J$ exchange—i.e. breakdowns in the approximation of Eq. (4)—due to differences in the transient exchange response to the two simultaneous voltage pulses. Nonetheless, the residual error can likely be mitigated through pulse-shaping techniques and the reduced gate depths afforded by simultaneous exchange already confer a measurable advantage—one that is expected to become even more significant for two-qubit operations[37].

## Discussion

We have demonstrated high-fidelity quantum control of an AEON qubit that is operated using simultaneous 2-$J$ exchange pulses. The performance of the AEON qubit was validated through BRB, yielding an average Clifford gate fidelity $F_{C1} = 99.86\%$ and an average error per pulse of 0.076%. Measurements of 2-$J$ exchange oscillations ($N_{osc}$) consistently outperformed 1-$J$ exchange oscillations, suggesting that pulsing simultaneous exchange is less sensitive to charge noise and that the 2-$J$ gate fidelity is currently limited by differences in the transient response of the two exchange interactions. The remaining error could be mitigated in the future by temporal pulse shaping using optimal control protocols[45].

Our work can be extended in several fascinating directions. First, 2-$J$ exchange offers substantial reductions in gate depths for entangling operations, reducing the number of exchange pulses by well over an order of magnitude as compared to typical EO qubit entangling gates[23,24]. Therefore, demonstrating entanglement of two AEON qubits would be a milestone. Second, for single-qubit operation, 3-$J$ exchange enables the construction of a leakage-protected identity gate[23], which can further suppress leakage errors during idle. Lastly, simultaneous exchange could be utilized to implement more efficient native $i$-Toffoli gates in arrays of Loss-DiVincenzo single-spin qubits[46].

## Methods

### Virtual gates

Virtual gate voltages $\widetilde{V}$ are related to physical gate voltages $V$ through a compensation matrix $C$, such that $\widetilde{V} = CV$[28–30]. Ideally, the chemical potential $\epsilon_i$ is only affected by a voltage $V_{P_i}$ applied to the plunger gate $P_i$. In reality, voltages applied to neighboring gates will also affect $\epsilon_i$.

The virtual gates, as defined by $C$, are designed to compensate for this cross-coupling to first order. Explicitly, in this work, we used:

$$\begin{pmatrix} \widetilde{V}_{P_1} \\ \widetilde{V}_{P_2} \\ \widetilde{V}_{P_3} \\ \widetilde{V}_{X_{12}} \\ \widetilde{V}_{X_{13}} \\ \widetilde{V}_{X_{23}} \end{pmatrix} = \begin{pmatrix} 1 & 0.19 & 0.18 & 0.51 & 0.67 & 0.21 \\ -0.19 & 1 & 0.20 & 0.38 & 0.36 & 0.49 \\ 0.06 & 0.20 & 1 & 0.16 & 0.98 & 0.53 \\ 0 & 0 & 0 & 1 & 0 & 0 \\ 0 & 0 & 0 & 0 & 1 & 0 \\ 0 & 0 & 0 & 0 & 0 & 1 \end{pmatrix} \begin{pmatrix} V_{P_1} \\ V_{P_2} \\ V_{P_3} \\ V_{X_{12}} \\ V_{X_{13}} \\ V_{X_{23}} \end{pmatrix}.$$

(5)

Here, the matrix elements are defined as $C_{i,j} = (\partial \epsilon_i / \partial V_{P_j}) / (\partial \epsilon_i / \partial V_{P_i})$ with analogous definitions for exchange gates. We determine the values of the $C_{i,j}$ by tracking shifts of electron loading lines for each dot as a function of gate voltage.

In the Fig. 2 data, we also compensated for first-order cross-coupling between exchange gates using:

$$\begin{pmatrix} \widetilde{V}'_{X_{1,2}} \\ \widetilde{V}'_{X_{1,3}} \\ \widetilde{V}'_{X_{2,3}} \end{pmatrix} = \begin{pmatrix} 1 & d_{12,13} & d_{12,23} \\ d_{13,12} & 1 & d_{13,23} \\ d_{23,12} & d_{23,13} & 1 \end{pmatrix} \begin{pmatrix} \widetilde{V}_{X_{1,2}} \\ \widetilde{V}_{X_{1,3}} \\ \widetilde{V}_{X_{2,3}} \end{pmatrix}$$

(6)

$$= \begin{pmatrix} 1 & -0.08 & -0.08 \\ -0.24 & 1 & -0.18 \\ -0.15 & -0.19 & 1 \end{pmatrix} \begin{pmatrix} \widetilde{V}_{X_{1,2}} \\ \widetilde{V}_{X_{1,3}} \\ \widetilde{V}_{X_{2,3}} \end{pmatrix},$$

(7)

where $d_{ij,kl} = (\partial J_{i,j} / \partial \widetilde{V}_{X_{k,l}}) / (\partial J_{i,j} / \partial \widetilde{V}_{X_{i,j}})$. Here, the $d_{ij,kl}$ were determined by tracking the first exchange fringe in the fingerpinch plots as a function of neighboring exchange gate voltage near the center of the sweep. To simplify notation, we drop the superscripts in the labels of the virtual exchange gates in Fig. 2. We emphasize that exchange gate compensation was not used in any of the other data presented in this work (in particular, it was not applied for the calibration sweeps shown in Fig. 4).

## Data availability

The data generated in this study have been deposited in the database https://doi.org/10.6084/m9.figshare.31115749.

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

## Acknowledgements

We thank John B. Carpenter for assisting with the preparation of the figures, Quantum Machines for access to the hardware used to perform the experiments (QDAC-II and OPX+), and Minh Nguyen for logistical support. The sample used in this experiment was made by the HRL device fabrication team. Research was supported by Army Research Office (ARO) grants W911NF-24-1-0020 and W911NF-22-C-0002 awarded to J.R.P. The views and conclusions contained in this document are those of the authors and should not be interpreted as representing the official policies, either expressed or implied, of the ARO or the U.S. Government. The U.S. Government is authorized to reproduce and distribute reprints for Government purposes, notwithstanding any copyright notation herein.

## Author contributions

J.D.B. and E.A. conceptualized the experiment. J.D.B. designed the experiments and collected the data. J.D.B. and J.C.H. analyzed the data and prepared the figures with input from J.R.P. J.R.P. designed the device. J.R.P. and E.A. supervised the project. All authors contributed to the writing of the manuscript.

## Competing interests

J.R.P. has a significant financial interest in HRL Laboratories, LLC. The remaining authors declare no competing interests.
