## [Transparent Peer Review file · Nature Communications]

Demonstration of an always-on exchange-only spin qubit

Corresponding Author: Professor Jason Petta

Version 0:

Reviewer comments:

Reviewer #1

(Remarks to the Author)

This work experimentally demonstrates the always-on exchange-only (AEON) qubit, a variation of the exchange-only (EO) qubit encoded in a three-spin system. The AEON qubit exhibits two key features: (i) a double sweet spot (DSS) where the qubit is insensitive to detuning fluctuations up to first order, and (ii) the ability to tune multiple exchange interactions by controlling the barrier potential while remaining on the DSS, enabling the implementation of all gate operations within this regime.

In this work, the AEON qubit is realized in a triangular triple quantum dot system in isotopically enriched silicon.

A significant challenge in implementing the AEON qubit lies in the simultaneous calibration of multiple gate voltages to both locate the DSS and apply gate operations effectively. The authors address this by employing virtual gates and calibration schemes to identify the optimal gate voltages. The experimental results show that the AEON qubit performs slightly better when utilizing two simultaneous exchange interactions, as compared to operating with a single exchange interaction at a time.

While the performance improvement over conventional EO qubits is modest, this work demonstrates the calibration of multi-exchange operations, which could be highly beneficial in larger systems containing multiple AEON qubits.

This is an important experimental contribution, showcasing the AEON qubit for the first time, and providing a viable alternative to traditional EO qubits. I recommend its acceptance for publication in Nature Communications.

However, I would like the authors to address the following points:

The observed performance gain is modest. What do you identify as the main limiting factors? Would pulse-shape engineering, which ensures the Hamiltonian commutes at different times, potentially help to improve performance?

One of the key technical advancements in this work is the calibration scheme for controlling two exchange interactions (2-J), which appears to be more complex than the 1-J calibration. Could you comment on whether this complexity might present challenges in scaling up to larger systems with more quantum dots?

Reviewer #2

(Remarks to the Author)

Reviewer #3

(Remarks to the Author)

The authors of the manuscript show the formation of an AEON qubit in a series of three quantum dots and demonstrate the operation of the qubit at the DSS point. They examine two different modes of driving exchange oscillations between the

different dots in the qubit, with the key difference between the modes being the difference in the axes around which the qubit is being driven. Finally, the authors perform interleaved BRB of the exchange oscillations and demonstrate that high fidelity exchange oscillations are performed. In general, the manuscript is well-written, and the work is highly interesting. However, I would strongly suggest that the comments below are addressed before the manuscript is published, particularly surrounding the apparent contradiction between Fig.3 and the text.

To what extent have the authors characterised the position of the DSS? I assumed that the authors chose the DSS point to be where the differential detunings are zero, but do the authors expect any 2nd order terms to dominate there and based on the charge noise level, do the authors expect the DSS to shift slightly depending on the exact charge configuration?

I am confused by what appears to be a contradiction between Fig.3 and the text in the last paragraph of the quantum control section. In Fig.3, b) and c) are respectively 1-J and 2-J oscillations, clearly indicating that 2-J oscillations are worse than 1-J. But the text claims N_{osc} for 2-J is greater than 1-J, which is the opposite of what is shown in Fig.3. Then, the authors claim that operating at the DSS should yield similar N_{osc} for both regimes. Could the authors explain in greater detail why they believe this is the case? My naïve intuition is that the 2-J oscillations would be susceptible to noise against more axes and therefore, it makes sense that 2-J oscillations are performing worse. The higher values of N_{osc} for 2-J exchange is referenced again in the later section – and it is unclear if then Fig.3 is labelled wrong. I am concerned with the inconsistencies in the explanations given the inconsistency in the way the results are described.

On that note, do the authors find that interleaved BRB suffer from the same problems that interleaved RB has, which is that there can be decoupling effects introduced in the reference sequences, which in principle would overestimate the fidelities of the gates being interleaved and could potentially give fidelities of $> 100\%$ if decoupling is particularly good. Have the authors considered other forms of benchmarks and explore how they compared relative to interleaved BRB?

What are the steps needed to go from these intra-dot exchange oscillations to performing a single qubit gate? I am assuming that 1-J exchange oscillations are sufficient and that going to 2-J and 3-J exchange is a matter of improving the protection against noise.

Also, can the authors comment more on the leakage observed in this system, i.e., how significant it is and whether the leakage observed was due to readout or during operation? This question stems from just the basic understanding that leakage is a non-negligible source of noise and I am interested in how impactful it is here, beyond just the number being quoted.

Version 1:

Reviewer comments:

Reviewer #1

(Remarks to the Author)

The authors addressed all referees' comments satisfactorily, and I recommend its acceptance for publication.

Reviewer #2

(Remarks to the Author)

Reviewer #3

(Remarks to the Author)

I think the authors have sufficiently addressed my comments and I support publication of the manuscript.

We thank all the reviewers for their time, careful evaluations, and insightful feedback, which we have incorporated into an improved manuscript. Below, we address each of the reviewer's comments point by point and summarize the resulting changes made to the manuscript.

Reviewer 1

Comment 1:

The observed performance gain is modest. What do you identify as the main limiting factors? Would pulse-shape engineering, which ensures the Hamiltonian commutes at different times, potentially help to improve performance?

Response:

The modest performance gain of 2-J over 1-J Clifford gates is primarily attributed to reduced gate depths. On average, a 2-J Clifford requires 1.9 exchange pulses, compared to 2.7 exchange pulses for a 1-J Clifford. More broadly, for arbitrary single-qubit unitaries we expect roughly a twofold reduction in gate depth when using 2-J exchange rather than 1-J exchange (Ref. 22). However, blind randomized benchmarking (BRB) measurements show that the average error per 2-J exchange pulse (0.076%) is actually higher than the best average error per 1-J pulse (0.060%). Some care is warranted in interpreting this comparison: a complete 1-J Clifford set can be compiled using only two exchange axes, whereas implementing arbitrary 2-J rotations necessarily involves all three. The elevated 2-J error could therefore partially reflect poorer performance along the third axis.

To understand the discrepancy more systematically, it is useful to distinguish between noise-driven errors and coherent control errors. Our measurements indicate that 2-J exchange is *less* sensitive than 1-J exchange to the two dominant noise mechanisms: charge noise, as quantified by N_{osc} and hyperfine-induced fluctuations, as inferred from the leakage rate extracted from BRB. This suggests that the reduced per-pulse fidelity of 2-J exchange is not noise-limited but instead arises from coherent errors. While we cannot rule out contributions from imperfect calibration or increased 2-J susceptibility to overlap of sequentially applied exchange pulses (see Refs. 18 and 19), we suspect that the dominant effect is transient pulse distortion: during the rising and falling edges of the pulse, the ratio of the two simultaneous exchange couplings deviates from its intended value, violating the idealized rotation model (Eq. 4 of the main text).

In future work, we plan to investigate these transient effects more directly using advanced characterization tools such as gate-set tomography, and to explore pulse-shape engineering—guided by optimal-control methods—as a path toward mitigating these coherent errors.

Changes made in response to comment:

We modified paragraph 2, first column, fifth page to include a discussion of this point:

Although 2-J BRB outperforms 1-J BRB, this advantage primarily arises from its shorter gate depths: a 2-J single-qubit Clifford gate requires an average of 1.9 exchange pulses,

compared to 2.7 pulses for a 1-J Clifford gate. When this difference is taken into account, the total 2-J error per exchange pulse ($\epsilon_{pp} \approx 0.076\%$) is in fact higher than that of the best-performing 1-J gates ($\epsilon_{pp} \approx 0.060\%$). This outcome is somewhat unexpected given the larger N_{osc} and lower leakage rates observed for 2-J exchange, which imply a reduced sensitivity to charge and hyperfine noise --- the dominant noise sources in EO qubits [18, 19]. This discrepancy suggests an increased contribution from coherent errors during 2-J pulsing, possibly due to imperfect calibration or enhanced sensitivity to pulse overlap [18], but we suspect the dominant contribution stems from the non-commutativity of 2-J exchange --- i.e. breakdowns in the approximation of Eq. 4 --- due to differences in the transient exchange response to the two simultaneous voltage pulses. Nonetheless, the residual error can likely be mitigated through pulse-shaping techniques and the reduced gate depths afforded by simultaneous exchange already confer a measurable advantage --- one that is expected to become even more significant for two-qubit operations [37].

Comment 2:

One of the key technical advancements in this work is the calibration scheme for controlling two exchange interactions (2-J), which appears to be more complex than the 1-J calibration. Could you comment on whether this complexity might present challenges in scaling up to larger systems with more quantum dots?

Response:

We expect that the extent to which the calibration scheme poses challenges for scaling to multi-qubit systems will depend strongly on the specific device geometry (e.g. linear, square 2D array, 2D hex array). In a close-packed geometry, cross-coupling between neighboring gate electrodes may be substantial, potentially necessitating modifications to the calibration protocol. In contrast, in a more conservative architecture -- where adjacent qubits interact only through a single exchange gate electrode -- we anticipate that the present protocol should generalize without significant changes.

More broadly, as in the one-dimensional analogue (Ref. 42), our calibration procedure exhibits Heisenberg scaling with respect to the sequence length N , making it both efficient and well-suited for multi-qubit extensions. In principle, calibration of different qubits could also be carried out in parallel, with the possible exception of immediately adjacent qubits where cross-talk effects are strongest.

Changes made in response to comment:

We modified paragraph 2, second column, fourth page to include a discussion of this point:

A specific rotation $\hat{R}_\varphi(\theta)$ is calibrated by sweeping the amplitudes of the exchange gate pulses to produce a two-dimensional map of F. An example sweep for a 2-J π -rotation about the \hat{z} -axis is shown in Fig. 4, where a pre-calibrated 2-J π -rotation about the \hat{x} -axis

is used as $\hat{R}_\eta(\chi)$ in the construction of \hat{U}_{ax} . As the exchange voltages are varied, F oscillates periodically, producing a series of interference peaks whose frequency --- and therefore sensitivity --- increases with the sequence length N . Only the central peak corresponds to the optimal calibration condition $\varphi = \varphi^*$ and $\theta = \theta^*$, which we track by performing successive sweeps with increasing values of $N=1,2,4,\dots$, continuing until reductions in the signal-to-noise ratio prevents scaling beyond $N = 24$. This calibration procedure is effectively a two-dimensional extension of the robust phase estimation protocol of Kimmel *et al.* [41] and similarly benefits from a Heisenberg scaling with respect to sequence length N , making it well suited for larger multi-qubit systems. In practice, we perform low-resolution sweeps for intermediate values of N and use a generic peak-finding algorithm to locate the central maximum, reserving a high-resolution sweep and analytic fit (see supplementary information) for the final $N=24$ iteration. Using this procedure, a full calibration of a single 2-J rotation takes approximately five minutes.

Reviewer 2:

Comment 1:

Although they claimed the advantage for the AEON qubit, this fidelity is comparable to the work in Ref [18], and is still lower than 99.9%. This means the AEON scheme did not work that well. Meanwhile, two-qubit control is also lack in their manuscript. More importantly, the recent reported single-qubit control for spin qubits in silicon has reached fidelity surpassing 99.99% (arXiv:2507.11918, 2025).

Response:

We agree that the improvement of the AEON qubit over the EO qubit measured in the same device (and similar devices) is modest. However, we believe that this modest improvement is already quite interesting given the large gains (over an order of magnitude shorter gate operations) expected when applied to large multi-qubit systems. Meanwhile, our work answers three outstanding questions regarding simultaneous exchange control needed for AEON qubit implementation:

1. Will charge noise be too costly for effectively implementing AEON qubits?

Answer: By operating at a DSS, we are able to realize simultaneous exchange oscillations with quality factors that actually exceed those of non-simultaneous exchange operated at an analogous symmetric operating point.

2. Can quantum gates be effectively calibrated given the complicated cross-coupling of the exchange interaction?

Answer: We have demonstrated a calibration protocol that is robust to cross-coupling between exchange interactions and that is quite efficient, achieving Heisenberg scaling with respect to the number of applications of the target gate.

More generally, this protocol represents a two-dimensional extension of an existing protocol based on robust phase estimation that is useful for cases where gate parameters must be calibrated simultaneously.

3. Will the error due to imperfect and slightly different pulsing between the different exchange interactions be a fundamental limitation?

Answer: While this may be a dominant error source for the AEON qubit, our results provide an upper-bound on the size of this error and we find that it is still low enough that the AEON qubit provides a net-benefit relative to the conventional EO qubit.

Lastly, we would like to emphasize that the 99.99% single qubit gate fidelities reported by the reviewer are associated with Loss-DiVincenzo (LD) single spin qubits, not exchange only spin qubits. LD spin qubits generally require the incorporation of a micromagnet for the efficient driving of electric dipole spin resonance. It is unlikely that micromagnets can be incorporated into industrially fabricated spin qubit devices with multiple back-end-of-line (BEOL) layers that increase the distance between the micromagnet and spins, thereby reducing Rabi frequencies. The investigation of other spin qubit modalities, for example exchange only spin qubits and singlet triplet spin qubits, is therefore warranted. Exchange only spin qubits, in particular, benefit from short gate times that are enabled by large exchange couplings. In contrast, slow 1 – 3 MHz gate speeds were required to achieve the high fidelities reported in 2507.11918. A singular focus on one performance metric, in this case the LD single qubit gate fidelity, is misguided, as would be a comparison with much different physical systems (e.g. trapped ions, where the fidelities are orders of magnitude higher than any solid state qubit).

Comment 2:

The authors are encouraged to introduce how to implement EO qubit and AEON in detail. Particularly, they should compare how difference between the two qubits. Also, the definition of the so-called 1-J and 2-J exchange controls in the manuscript is not so clear.

Response:

We define the technical difference between the EO/AEON qubits in the second paragraph, first column, second page: **EO or AEON qubit control is then distinguished by the restriction of \hat{H} to 1-J or 2-J exchange, respectively.** But agree that it should be further clarified given its central importance to our results.

Changes made in response to comment:

We have added several sentences to the second paragraph, first column of the second page to make the distinction between an EO qubit and AEON qubit clearer:

EO or AEON qubit control is then distinguished by the restriction of \hat{H} to 1-J or 2-J exchange, respectively. **That is, an AEON qubit is distinguished from an EO qubit by its mode of operation. EO qubit control consists of serial pulsing of exchange where only a single exchange interaction is active at a time, while AEON control consists of simultaneous exchange where at least two exchange interactions are active at a time (2-J exchange).**

Both the EO and AEON qubit are encoded identically in the collective three-electron spin state of the array, which occupies an eight-dimensional Hilbert space [15]...

Comment 3:

It is not so clear to the reader that how to derive Eq.2 from Eq.1. Can the authors add some derivation in detail in the appendix?

Response:

A detailed derivation of Eq. 2 from Eq. 1 is provided in section 1 of the supplemental material of Ref. 18.

Changes made in response to comment:

We have added a sentence immediately following the introduction of Eq. 2 pointing to the derivation in Ref. 18:

A detailed derivation of Eq. 2 from Eq. 1 is provided in the supplemental material of Ref. [18].

Reviewer 3:

Comment 1:

To what extent have the authors characterised the position of the DSS? I assumed that the authors chose the DSS point to be where the differential detunings are zero, but do the authors expect any 2nd order terms to dominate there and based on the charge noise level, do the authors expect the DSS to shift slightly depending on the exact charge configuration?

Response:

We have not characterized the location of the DSS beyond performing sweeps of the type shown in Fig. 3a of the main text, from which we estimate the DSS position by eye. For each 2-J rotation axis, we select a single DSS point corresponding to a π -rotation (for a 10 ns pulse) and use this same point for all other rotation angles. Because we employ virtual gates that compensate the linear cross-capacitance of the exchange gates onto the plunger gates, we find that the shift in the DSS position across different rotation angles is negligible. Long term drifts of the charge configuration and thus DSS due to charge noise are also possible, but these could be handled with intermittent recalibration sequences. A more detailed investigation -- such as mapping the insensitivity region as a function of plunger-gate voltages, in analogy with Ref. 41 - would likely yield valuable insights, and we plan to pursue this in future work.

Regarding higher-order terms at the DSS, Ref. 21 provides a detailed theoretical analysis within the context of the Fermi–Hubbard model. While this work shows that exchange does acquire a second-order dependence on differential detunings, it also demonstrates a first-order dependence on tunnel couplings. We expect this latter channel to dominate the impact of charge noise on exchange coherence, analogous to the behavior seen in the non-simultaneous exchange case discussed in Ref. 41.

Changes made in response to comment:

We have added the following sentence at the end of paragraph 2, second column, page 3:

Although exchange energies retain a second-order dependence on chemical potentials at the DSS [21], these contributions are expected to be negligible compared to the dominant first-order dependence on the interdot tunnel couplings, as is the case for 1-J exchange operated at the analogous symmetric operating point [41].

Comment 2:

I am confused by what appears to be a contradiction between Fig.3 and the text in the last paragraph of the quantum control section. In Fig.3, b) and c) are respectively 1-J and 2-J oscillations, clearly indicating that 2-J oscillations are worse than 1-J. But the text claims N_{osc} for 2-J is greater than 1-J, which is the opposite of what is shown in Fig.3.

Response:

The confusion likely arises because Fig. 3b displays 2-J exchange oscillations while Fig. 3c displays 1-J exchange oscillations, which is inconsistent with the ordering used in the original caption text.

Changes made in response to comment:

We have updated the wording in the Fig. 3 caption to read “**Measured 2- and 1-J exchange oscillations...**” so that it matches the ordering of the sub-figures. We have also added labels to

panels (b) and (c) to clearly indicate which data correspond to 1-J exchange oscillations and which correspond to 2-J exchange oscillations (see below).

Comment 3:

Then, the authors claim that operating at the DSS should yield similar N_{osc} for both regimes. Could the authors explain in greater detail why they believe this is the case?

Response:

We operate both 1- and 2-J exchange at “sweet spots” that are first-order insensitive to the chemical potentials. For 2-J exchange this is the DSS, for 1-J exchange it is typically referred to as the symmetric operating point (SOP) (Ref. 41). At both types of sweet spots, the dominant channel through which charge noise couples into the system is modulation of the interdot tunnel couplings. Granted there is only a single interdot tunnel coupling involved in 1-J exchange and two interdot tunnel couplings involved in 2-J exchange, but a significant portion of their energies interacts destructively. This can be seen quantitatively by computing the derivatives $\frac{d\Omega}{dV_i}$ of Eq. 2 of the main text for all gate electrode voltages V_i . Here, Ω is as defined on page 2 of the main text and describes the frequency of exchange oscillations, so that these derivatives describe the sensitivity of this frequency to gate-referred voltage fluctuations.

Changes made in response to comment:

That being said, these derivatives depend on a model of exchange energy as a function of gate voltage, which may vary significantly between the 1- and 2-J regimes. We have therefore rewritten the ending of the first paragraph, first column of page 4 to reduce the strength of our claim:

Operating at the DSS, we characterize coherence by measuring the qubit's time-evolution during 2-J exchange pulses, as plotted in Fig. 3(b). We quantify coherence by the number of exchange oscillations N_{osc} that occur before the amplitude decays to $1/e$ of its initial value. For comparison, we also plot the measured time-evolution for 1-J exchange operated at symmetric operating points in Fig. 3(c) (see Supplementary Information for

1-J fingerprint plots and associated operating points). We find that the N_{osc} for 2-J exchange is generally greater than for 1-J exchange, despite both regimes operating at sweet spots with vanishing first-order sensitivity to chemical potential fluctuations. More detailed device modeling may be helpful for understanding the coherence enhancement during 2-J operation.

Comment 4:

Do the authors find that interleaved BRB suffer from the same problems that interleaved RB has, which is that there can be decoupling effects introduced in the reference sequences, which in principle would overestimate the fidelities of the gates being interleaved and could potentially give fidelities of $> 100\%$ if decoupling is particularly good. Have the authors considered other forms of benchmarks and explore how they compared relative to interleaved BRB?

Response:

We indeed find that the leakage rates extracted from 2-J interleaved BRB are typically negative, which we attribute to a dynamical decoupling effect (see Table 1 in the Supplementary Information). For this reason, we choose to emphasize standard BRB in the main text, comparing sequences composed solely of 2-J exchange pulses with those composed solely of 1-J pulses. In future work, we plan to perform a more detailed error analysis of 2-J exchange using gate-set tomography, although this protocol will require modifications to properly account for leakage states in the EO qubit encoding.

Comment 4:

What are the steps needed to go from these intra-dot exchange oscillations to performing a single qubit gate? I am assuming that 1-J exchange oscillations are sufficient and that going to 2-J and 3-J exchange is a matter of improving the protection against noise.

Response:

The key steps required to transition from intra-dot exchange oscillations to fully calibrated single-qubit gates are outlined at the beginning of the *Gate Calibration and Benchmarking* section in the main text. In particular, we introduce an efficient calibration procedure based on the robust phase-estimation protocol of Kimmel *et al.* (Ref. 42), which enables the simultaneous calibration of both the rotation angle and rotation axis of a 2-J rotation. This is essential because these quantities are intrinsically coupled at the Hamiltonian level (Eq. 4) and are further coupled due to exchange cross-coupling. In this way, calibrating 2- and 3-J exchange is fundamentally different than calibrating 1-J exchange, which can be achieved directly using 1-J exchange oscillations.

Comment 5:

Also, can the authors comment more on the leakage observed in this system, i.e., how significant it is and whether the leakage observed was due to readout or during operation?

Response:

The measured leakage error of 0.029% for the 1-J EO qubit implementation is comparable to the value measured in similar SLEDGE devices (see Refs. 3, 19, 26). For the 2-J AEON qubit implementation, the measured leakage error of 0.015% is about half of this value, which is consistent with the interleaved BRB results and the leakage-protected nature of 2-J exchange (effectively, the AEON qubit is only exposed to significant leakage during idle periods between exchange pulses, which make up about half of the quantum circuit).

The leakage error measured using BRB corresponds only to the leakage during gate operations. The leakage error during readout is subsumed into the SPAM fidelity (essentially given by the contrast of the two curves in Fig. 5 extrapolated to 0 Clifford operations) and is comparable between AEON and EO qubit implementations (as we should expect, since readout is performed identically in both cases).

Changes made in response to comment:

We have added a statement in the third paragraph, first column, of the fifth page to give context to the leakage error:

Additionally, the 2-J leakage error per Clifford, 0.015%, is approximately half that of the average 1-J leakage error per Clifford, 0.029%, **with the 1-J leakage error being comparable to the value measured in similar devices [3, 19, 26].**

Additional Changes:

- There was a typo in the text of Fig. 1(d). We changed $J_{k,l} = 100$ MHz to $J_{k,l} = 2\pi \times 100$ MHz.
- There was a typo in Fig. 1(a). The charge configurations were mislabeled. We changed (clockwise from upper left) 102, 210, 012, 120 to 120, 012, 210, 102.
- We added a short statement in the conclusion to emphasize the expected improvement of entangling operations for the AEON versus EO qubit:

First, 2-J exchange offers substantial reductions in gate depths for entangling operations, reducing the number of exchange pulses by well over an order of magnitude **as compared to typical EO qubit entangling gates [23, 24].**

Resubmission of Broz *et al.*, “Demonstration of an always-on exchange-only spin qubit,” NCOMMS-25-68445-T

Dear Editors,

Thank you for sending our manuscript out for a second round of review. Reviewers 1 and 3 now completely support publication in Nature Communications. We are pleased that Reviewer 2 carefully considered our response and appreciates our perspective, stating “I agree with this perspective, which addresses the point I was most concerned about and directly defines the significance of the AEON qubit.” Reviewer 2 posed a final technical question which we address in the response below.

Best regards,
Jason Petta

Reviewer comments are in blue and our responses are in black.

Reviewer 2:

The revised manuscript has clarified the distinction between AEON and LD qubits, emphasizing that their present fidelities cannot be compared in a simplistic manner and that additional factors such as controllability and scalability must also be taken into account. I agree with this perspective, which addresses the point I was most concerned about and directly defines the significance of the AEON qubit. However, a key issue that is central to the interpretation of the work is still not fully resolved, namely, the physical origin of the performance advantage. The current version continues to emphasize the higher Clifford fidelity of AEON gates. However, the data show that the per-pulse error of 2-J operations (0.076%) is larger than that of 1-J operations (0.060%). This indicates that the fidelity improvement arises primarily from reduced pulse depth rather than intrinsically improved noise resilience. The manuscript should more clearly state this distinction and avoid implying a fundamental superiority of AEON control unless supported by additional evidence. A brief comparison at fixed total gate time, or a concise filter-function or noise-spectral discussion, would be sufficient to clarify this point.

Response:

We appreciate the Reviewer’s consideration of our stance that it is not practical to compare the performance of different quantum computing platforms using a single metric such as gate fidelity. Other factors, such as scalability, gate speed, and connectivity are just as important in the overall performance of a quantum processor. We are pleased that the Reviewer’s most serious concern has been addressed satisfactorily.

The Reviewer has also commented on the general interpretation of our work. The main contribution of the manuscript is the first demonstration of an AEON qubit. Figure 2 shows that we can control two exchange couplings simultaneously. Figure 3(a) further shows that fine tuning to a double sweet spot is possible. One important unanticipated result relates to the data sets presented in Figs. 3(a,b). We expected N_{osc} to be smaller for 2-J control as compared with 1-J control simply because 2-J control would be susceptible to charge noise on two exchange couplings. Instead, we find that N_{osc} for 2-J control is significantly larger than for 1-J. This result has been reproduced across several triangle devices

and is believed to be general. We objectively state in the manuscript that “More detailed device modeling may be helpful for understanding the coherence enhancement during 2- J operation.” In terms of the interpretation of blind randomized benchmarking (BRB) performance, we reproduce directly from the main text “Although 2- J BRB outperforms 1- J BRB, this advantage *primarily arises from its shorter gate depths*: a 2- J single-qubit Clifford gate requires an average of 1.9 exchange pulses, compared to 2.7 pulses for a 1- J Clifford gate.” We believe this statement is clear as is. We anticipate that our experimental results will spur on the development of sophisticated noise models. We also believe that such work is best performed by theoretical device modeling groups.

The manuscript “Demonstration of an always-on exchange-only spin qubit” by Joseph D. Broz et.al demonstrates high-fidelity control of an always-on exchange-only (AEON) qubit. The control is implemented via simultaneous exchange pulses in a triangular quantum dot (QD) array, and the achieved averaged fidelity is 99.86%. Although they claimed the advantage for the AEON qubit, this fidelity is comparable to the work in Ref [18], and is still lower than 99.9%. This means the AEON scheme did not work that well. Meanwhile, two-qubit control is also lack in their manuscript. More importantly, the recent reported single-qubit control for spin qubits in silicon has reached fidelity surpassing 99.99% (arXiv:2507.11918, 2025). Therefore, I think the AEON scheme is technically sound, however, I do not believe it to be of sufficiently high significance to warrant publication in Nature communications.

Also, I have some minor suggestions:

1. The authors are encouraged to introduce how to implement EO qubit and AEON in detail. Particularly, they should compare how difference between the two qubits. Also, the definition of the so-called 1-J and 2-J exchange controls in the manuscript is not so clear.
2. It is not so clear to the reader that how to derive Eq.2 from Eq.1. Can the authors add some derivation in detail in the appendix?

The revised manuscript has clarified the distinction between AEON and LD qubits, emphasizing that their present fidelities cannot be compared in a simplistic manner and that additional factors such as controllability and scalability must also be taken into account. I agree with this perspective, which addresses the point I was most concerned about and directly defines the significance of the AEON qubit. However, a key issue that is central to the interpretation of the work is still not fully resolved, namely, the **physical origin of the performance advantage**. The current version continues to emphasize the higher Clifford fidelity of AEON gates. However, the data show that the per-pulse error of 2-J operations (0.076%) is larger than that of 1-J operations (0.060%). This indicates that the fidelity improvement arises primarily from reduced pulse depth rather than intrinsically improved noise resilience. The manuscript should more clearly state this distinction and avoid implying a fundamental superiority of AEON control unless supported by additional evidence. A brief comparison at fixed total gate time, or a concise filter-function or noise-spectral discussion, would be sufficient to clarify this point.